# Ubiquitin-protein ligase *Ubr5* cooperates with hedgehog signalling to promote skeletal tissue homeostasis

David Mellis[1☯], Katherine A. Staines[2☯], Silvia Peluso[3], Ioanna Ch. Georgiou[4], Natalie Dora[3], Malgorzata Kubiak[1], Rob van't Hof[5¤a], Michela Grillo[1], Colin Farquharson[6], Elaine Kinsella[1], Anna Thornburn[3], Stuart H. Ralston[5], Donald M. Salter[5], Natalia A. Riobo-Del Galdo[4‡], Robert E. Hill[3‡*], Mark Ditzel[1¤b‡]

1 Edinburgh CRUK Cancer Research Centre, MRC Institute for Genetics and Molecular Medicine, University of Edinburgh, Edinburgh, United Kingdom, 2 School of Pharmacy and Biomolecular Sciences, University of Brighton, Brighton, United Kingdom, 3 MRC Human Genetics Unit, MRC Institute for Genetics and Molecular Medicine, University of Edinburgh, Edinburgh, United Kingdom, 4 Leeds Institute of Medical Research and School of Molecular and Cellular Biology, University of Leeds, Leeds, United Kingdom, 5 Centre for Genomic and Experimental Medicine, MRC Institute for Genetics and Molecular Medicine, University of Edinburgh, Edinburgh, United Kingdom, 6 Roslin Institute and R(D)SVS, The University of Edinburgh, Edinburgh, United Kingdom

☯ These authors contributed equally to this work.
¤a Current address: Institute of Life Course and Medical Sciences, University of Liverpool, Liverpool, United Kingdom
¤b Current address: Exscientia Ltd., Dundee One, Dundee, United Kingdom
‡ NAR-DG, REH and MD also contributed equally to this work.
* bob.hill@igmm.ed.ac.uk

**Data Availability Statement:** All relevant data are within the manuscript and its Supporting Information files.

## Abstract

Mammalian Hedgehog (HH) signalling pathway plays an essential role in tissue homeostasis and its deregulation is linked to rheumatological disorders. UBR5 is the mammalian homologue of the E3 ubiquitin-protein ligase Hyd, a negative regulator of the Hh-pathway in *Drosophila*. To investigate a possible role of UBR5 in regulation of the musculoskeletal system through modulation of mammalian HH signaling, we created a mouse model for specific loss of *Ubr5* function in limb bud mesenchyme. Our findings revealed a role for UBR5 in maintaining cartilage homeostasis and suppressing metaplasia. *Ubr5* loss of function resulted in progressive and dramatic articular cartilage degradation, enlarged, abnormally shaped sesamoid bones and extensive heterotopic tissue metaplasia linked to calcification of tendons and ossification of synovium. Genetic suppression of smoothened (*Smo*)**,** a key mediator of HH signalling, dramatically enhanced the *Ubr5* mutant phenotype. Analysis of HH signalling in both mouse and cell model systems revealed that loss of *Ubr5* stimulated canonical HH-signalling while also increasing PKA activity. In addition, human osteoarthritic samples revealed similar correlations between *UBR5* expression, canonical HH signalling and PKA activity markers. Our studies identified a crucial function for the *Ubr5* gene in the maintenance of skeletal tissue homeostasis and an unexpected mode of regulation of the HH signalling pathway.

**Funding:** MD is supported by a University of Edinburgh Chancellor's Fellowship and funding from a Carnegie Research Incentive Grant (70356); BH by an MRC University Unit grant MM_UU_00007/8; NARD-G by the NIH (1R01GM088256) and a BBSRC project grant (BB/S01716X/1); KS by the MRC (MR/R022240/2) and CF by BBSRC through an Institute Strategic Programme Grant Funding (BB/J004316/1). The funders had no role in study design, data collection and analysis, decision to publish, or preparation of the manuscript.

**Competing interests:** The authors have declared that no competing interests exist.

## Author summary

Ubiquitin ligases modify proteins post-translationally which is essential for a variety of cellular processes. UBR5 is an E3 ubiquitin ligase and in *Drosophila* is a regulator of Hedgehog signaling. In mammals, the Hedgehog (HH) signalling pathway, among many other roles, plays an essential role in tissue maintenance, a process called homeostasis. A murine genetic system was developed to specifically eliminate UBR5 function from embryonic limb tissue that subsequently forms bone and connective tissue (ligaments and tendons). This approach revealed that UBR5 operates as a potent suppressor of excessive growth of normal cartilage and bone and prevents formation of bone in ectopic sites in connective tissue near the knees and ankle joints. In contrast to abnormal growth, UBR5 inhibits degradation of the articular cartilage that cushions the knee joint leading to extensive exposure of underlying bone. Furthermore, Ubr5 interacts with smoothened, a component of the HH pathway, identifying UBR5 as a regulator of mammalian HH signaling in the postnatal musculoskeletal system. In summary, this work shows that UBR5 interacts with the HH pathway to regulate skeletal homeostasis in and around joints of the legs and identifies targets that may be harnessed for biomedical engineering and clinical applications.

## Introduction

Ubiquitin ligases target proteins for ubiquitination which can modulate protein function by regulating protein degradaton, protein–protein interactions, and protein localization [1–4], and thus, provide important post-translational mechanisms essential for a variety of cellular processes. The *Drosophila* homologue of the mammalian *Ubiquitin Protein Ligase E3 Component N-Recognin 5* (UBR5), designated as *hyperplastic discs* (Hyd), was originally identified as a Drosophila tumor suppressor protein [5–7] and regulator of Hedgehog (HH) signalling [6]. Physical and genetic interactions with established components of the HH signalling pathway [7,8] strengthened Hyd's role as a regulator of HH signalling. We previously addressed a possible conserved role for UBR5 in HH-mediated processes in mice [9]. Although no overt effects were seen in patterning of the developing limb bud in mouse embryogenesis; here, we show that the coordinated action of Ubr5 with HH signalling is crucial to maintain skeletal tissue homeostasis associated with the appendicular skeleton postnatally and in adult mice.

HH signalling regulates cell processes that are critical for skeletal tissue development, growth and homeostasis [10]. Two HH ligands, Sonic- and Indian-Hedgehog (SHH and IHH, respectively) are widely expressed and function as extracellular signalling molecules that bind to cells expressing HH receptors such as patched-1 (PTCH1). Binding to PTCH1 results in de-repression of the G protein-coupled receptor, smoothened (SMO), and activation of SMO-associated canonical and non-canonical signalling pathways [11–13]. Activation of the SMO-associated canonical pathway results in stimulation of GLI-mediated transcription and expression of crucial target genes [7]. Activation of the recently identified SMO-associated non-canonical pathway relies on SMO's GPCR activity [14,15] and results in inhibitory heterotrimeric G protein-mediated inhibition of adenylate cyclase and a concomitant reduction in cyclic AMP (cAMP) levels [14,16,17]. Although not yet experimentally addressed, non-canonical signalling may also contribute to many of the well-described roles for canonical HH signalling in normal skeletal formation, maturation and maintenance [10,18].

At birth, IHH is the ligand that drives HH signalling within the growing limbs. Expression of *Ihh* is localized to a zone of postmitotic, prehypertrophic chondrocytes immediately

adjacent to the zone of proliferating chondrocytes [18–20] and is essential for endochondral ossification but also induces osteoblast differentiation in the perichondrium [21]. Dysregulation of this signalling pathway is detrimental to musculoskeletal tissue homeostasis [22,23]. Notably, studies have shown that increased HH signalling can drive pathological ectopic cartilage and bone formation in soft tissues [10] through the process of heterotopic chondrogenesis and heterotopic ossification (HO) [24]. Upregulation of HH signalling is believed to contribute to the rare disorder, progressive osseous heteroplasia (POH), which includes in its phenotypic spectrum soft tissue ossification. POH is caused by loss-of-function of *GNAS*, a G protein alpha subunit and activator of adenylate cyclase. A murine model of POH demonstrated that increased HH signalling as a consequence of *GNAS* loss-of-function in mesenchymal limb progenitor cells drove heterotopic ossification [25]. Similarly, synovial chondromatosis, a disease resulting in ossification of synovial tissue is associated with increased canonical HH signalling [26]. However, in contrast with cartilage and bone gain, elevated HH signalling is also associated with the cartilage degradation and loss [27,28]. Hence, appropriate HH signalling is normally involved in the suppression of ectopic, and genesis and maintenance of normtopic, cartilage and bone.

Here, we show that the loss of *Ubr5* function in *Ubr5^mt^* mice resulted in diverse musculoskeletal defects including spontaneous, progressive and tissue-specific patterns of ectopic chondrogenesis and ossification as well as articular cartilage degeneration and shedding. Surprisingly, reducing SMO function in UBR5-deficient mice led to a dramatic reduction in the age of onset and increased severity of the *Ubr5^mt^* phenotype. These observations challenge the existing dogma by highlighting an important role for *Smo*, in the absence of UBR5, in suppressing, rather than promoting, ectopic chondrogenesis, tissue calcification/ossification and articular cartilage damage. We, therefore, reveal a previously unknown physiological role for *Ubr5* and highlight its genetic interaction with *Smo* in regulating cellular and tissue-homeostasis. These findings may influence current therapeutic approaches modulating HH signalling for the treatment of degenerative musculoskeletal conditions such as osteoarthritis and heterotopic ossification.

## Results

### Loss of *Ubr5* function causes skeletal heterotopias at 6 months

To overcome the embryonic lethality associated with germline mutant animals [29], we combined a *Ubr5* conditional loss-of-function gene trap (*Ubr5^gt^*) [9] with *Prx1-Cre* [30] (*Prx1-Cre; Ubr5^gt/gt^* animals henceforth, referred to as *Ubr5^mt^*) to ensure that adult tissues derived from early limb bud mesenchyme, predominantly bone and connective tissue, were *Ubr5* deficient. Since the HH pathway affects embryonic limb patterning and bone growth, the *Ubr5* deficient fetuses (at E15.5) were initially examined and bones and joints appeared to develop normally [9]. However, the HH pathway continues to function in postnatal bone growth and homeostasis [10] and thus, at approximately 6 months of age, we noticed that mice began to display defects in locomotion. Control animals normally remained supported by their hindlimbs ('sprung'), whereas, *Ubr5^mt^* animals rested their posteriors directly upon the floor ('squat') (S1A–S1C Fig). Considering the tissue targeted by the conditional mutation, the observed phenotype indicated a potential musculoskeletal system defect which prompted the examination of hindleg bone and joint structures.

At 6 months of age, X-ray imaging revealed that *Ubr5^mt^* animals exhibited abnormally shaped and/or ectopic signals around knee and ankle joints (S1D–S1G Fig). 3D micro-computed tomography (μCT) revealed that, whereas *Prx1-Cre* control joints appeared normal with no evidence of ectopic structures (Fig 1A), the knees of all *Ubr5^mt^* mice (n = 10) exhibited

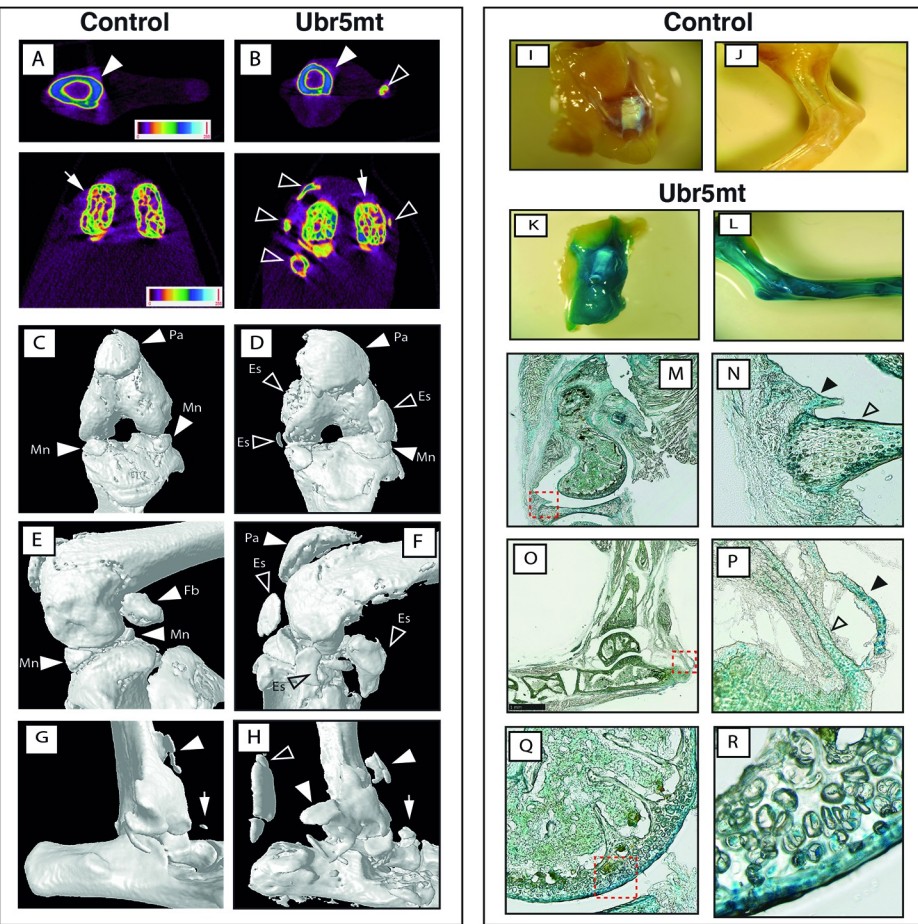

**Fig 1. *Ubr5^{mt}* animals exhibit multiple ectopic structures around the knee and ankle joints in *Ubr5* expressing tissue.** 6-month old control and *Ubr5^{mt}* animals were analysed by μCT. (A, B) Color-coded density maps revealed presence of ectopic, non-uniform density structures (open arrowheads) around the tibia (closed arrowheads) and femoral condyles (arrows). (C, D) Knee joints in ventral and (E, F) in medial (lower panels) aspect. Closed arrowheads indicate normal structures: the patella (Pa), menisci (Mn) and fabella (Fb). Open arrowheads indicate ectopic signals (Es) present in the *Ubr5^{mt}* knee joint. (G) Control ankle joints exhibit a signal extending from the ventral face of the tibia (closed arrowhead) and a small structure presumed to be a sesamoid bone (arrow). (H) Multiple ectopic signals were present around the *Ubr5^{mt}* ankle joint (arrowheads), including a large dorsally located and well-isolated structure in the location of the AT (open arrowhead). (I-R) 20-week-old *Prx1-Cre* control and *Ubr5^{mt}* ankle and knee joints were stained for β-gal activity. Whole mount knee (I, K) and ankle (J, L) are shown with subsequent sagittal sections for the knee (M) and the boxed area magnified in (N) shows the outer layer of the menisci (open arrowhead), and the adjacent synovium (closed arrowhead) stained positive for β-gal expression. Sagittal sections for the ankle are shown in (O) and magnified in (P) showing staining of the AT and superficial digital flexor tendon (open and closed arrowheads, respectively). (Q, R) Expression of β-gal in the articular cartilage of the knee and the red dashed box in Q shown at higher magnification in R.

isolated ectopic signals clearly separated from the adjacent femoral condyles and tibia (Fig 1B). Surface rendering of the μCT scans demonstrated that the array of knee-associated sesamoid bones (patella and fabella) and calcified menisci (Fig 1C and 1D) were abnormal. *Ubr5^{mt}* knees presented with large ectopic structures on all four faces of the knee joint, as well as enlarged and irregularly shaped fabella and patella sesamoid bones (Fig 1E and 1F). In addition, the *Ubr5^{mt}* animals exhibited multiple ectopic signals around the ankle joint (Fig 1G and 1H), with the most striking one appearing consistently on the dorsal side running parallel to the long axis of the tibia (Fig 1H, open arrows) associated with the Achilles tendon (AT). This

ectopic signal remained isolated from the calcaneus and tibia. Other ectopic structures included two ectopic U-shaped signals on the ventral and lateral sides of the tibia (Fig 1H).

Following *Prx1-Cre* mediated recombination of the *Ubr5^{st}* gene-trap construct, the reporter gene *lacZ* (encodes β-galactosidase [β-gal] activity) is activated and expressed under the influence of *Ubr5* gene regulators [9] acting as proxy for *Ubr5* expression. β-gal activity is initially detected in limb mesenchyme at early embryonic stages [9] and at the postnatal stages examined here, activity is restricted to limb tissue derived from this embryonic mesenchyme. Analysis of *lacZ* expression in 20 week-old control and *Ubr5^{mt}* mice indicated that *Ubr5* is expressed around the knee (Fig 1I and 1K) and ankle (Fig 1J and 1L) joints. Further analysis revealed strong β-gal activity around the periphery of the menisci and synovium (Fig 1 M and 1N). The ankle also revealed β-gal activity within the AT and superficial digital flexor tendon and in a large ectopic structure within the AT midbody (Fig 1O and 1P). Thus, the tissues that exhibit Cre-mediated expression of the *lacZ* gene are affected in the mutant phenotype.

## *Ubr5^{mt}*-associated ectopic structures exhibit chondrogenesis and calcification

The morphology of these ectopic structures around the knee (see Fig 1D and 1F) and ankle (see Fig 1H) joints were further investigated to determine the cellular composition and possible derivation of these ectopias. As shown by μCT, both knee (Fig 2A and 2B) and ankle (Fig 2D and 2E) ectopic structures harbored X-ray dense internal structures indicative of bone. Accordingly, in these joints (Fig 2C and 2F) von Kossa staining, which is widely used to detect abnormal calcium deposits, revealed staining in these ectopic structures. Von Kossa staining in the *Ubr5^{mt}* knee joints revealed positive stained structures within the synovium deep in the patellar tendon (Fig 2C) and near the ankle joint staining within the AT (Fig 2F). In addition, in the control AT, the expected ordered stacking of tenocytes along the anterior-posterior axis (Fig 2G) and an absence of toluidine blue staining associated with proteoglycans (Fig 2H) was observed. In contrast, these regions of the *Ubr5^{mt}* AT were devoid of tenocytes, which were replaced by long columns of proteoglycan-expressing hypertrophic chondrocytes (Fig 2I and 2J). Histological analysis of control knee joints revealed a synoviocyte-rich intimal layer of the synovium (Fig 2K), whereas *Ubr5^{mt}* knee joints exhibited bone- (Fig 2L) and cartilage-like (Fig 2M) ectopic structures. The observed mutant morphology was consistent with the formation of ectopic bone around the knee and in the AT, which was further associated with the presence of ectopic chondrocytes and the production of extracellular matrix.Thus, an abnormal phenotype consisting of ectopic chondrogenesis, calcification and ossification (hereafter, referred as ECCO) of the synovium and tendons in *Ubr5^{mt}* tissues was observed. We concluded that *Ubr5* normally functions to prevent spontaneous ectopic formation of chondrocytes in tissues and calcification and/or ossification in cartilage.

### Loss of *Ubr5* function causes articular cartilage degradation

μCT analysis of 6-month old control (Fig 3A) and *Ubr5^{mt}* (Fig 3D) knee revealed the previously described overt morphological differences (see above), but further analysis at the surface of the joints, showed a significantly increased volume of high subchondral bone density (designated by red in Fig 3B, 3C, 3E and 3F) in the mutant. The increase in volume as a percentage of total subchondral bone (depicted in S1H–S1J Fig) is quantified in Fig 3G. In addition, histological assessment showed a dramatic loss of articular cartilage (AC) from the lateral tibial and femoral surfaces of all *Ubr5^{mt}* knee joints assessed (Fig 3H and 3I); a condition not detected in any control mice at this stage. Further examination of the exposed subchondral bone in these *Ubr5^{mt}* mice revealed abnormal intermixed bone and cartilage within this region (Fig 3J and

## Ubr5mt

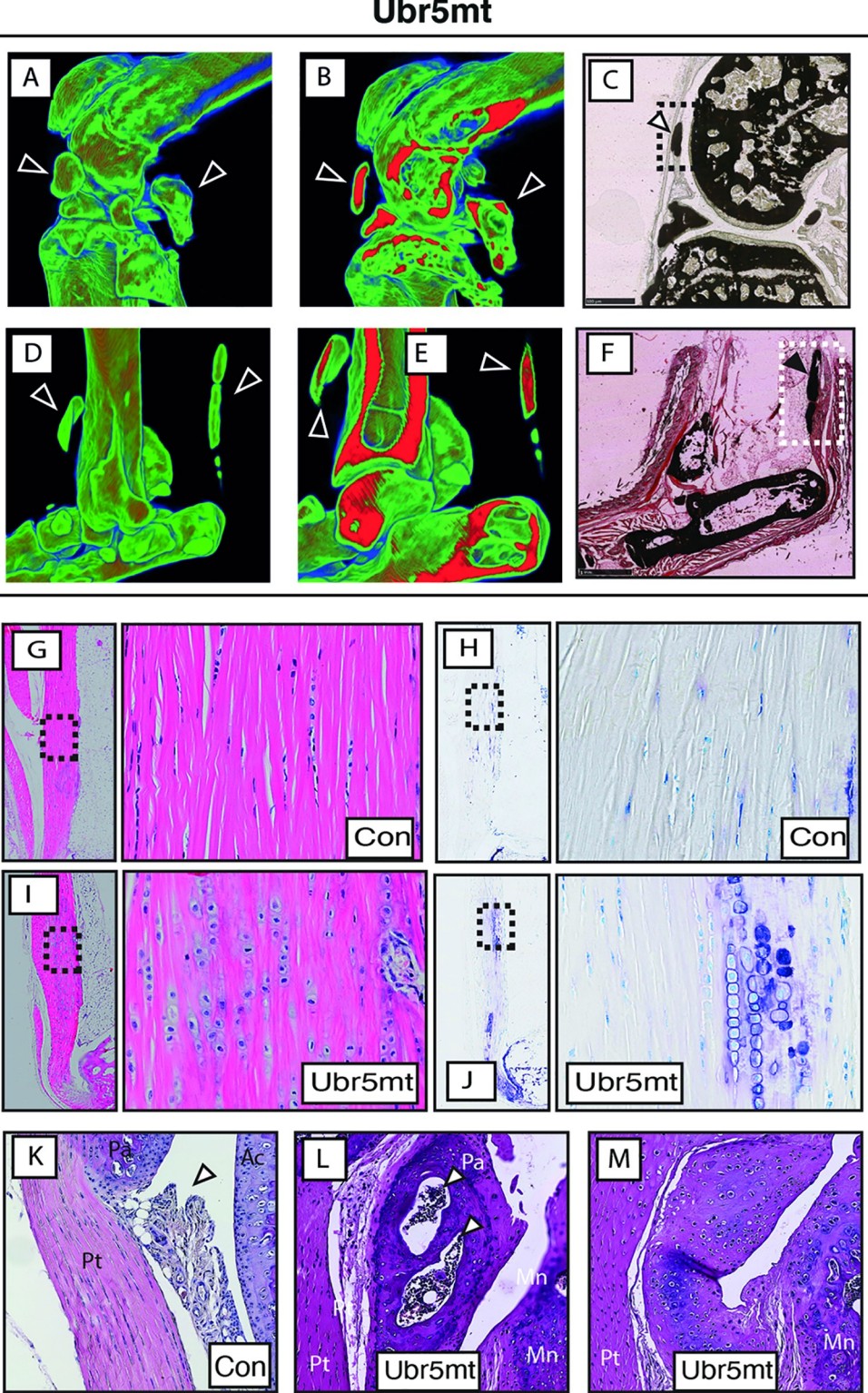

**Fig 2. *Ubr5^mt* limbs exhibit ectopic chondrogenesis, cartilage formation and calcification and ossification.** (A, B, D, E) Colour-coded X-ray density maps of volume rendered *Ubr5^mt* knee (A B) and ankle (D, E) joints. (B, E) show cross-sections through the joint to reveal the internal structure and density. Arrowheads indicate ectopic structures indicated in Fig 1F and 1H. Low density = blue and High density = red. Sagittal sections (C, F) from 20-week-old or 24-week-old animals are shown. (C, F) Von Kossa staining of *Ubr5^mt* knee (C) and ankle joints (F). (C) *Ubr5^mt* knee

joints revealed ectopic signals in the synovium (dashed box and arrowhead) lying within the synovium and under the patellar tendon. (F) *Ubr5^{mt}* ankle revealed ectopic signals in AT (dashed box and arrowhead) on the deep face of the AT. (G, I) H&E and (H, J) toluidine blue staining of the midbody of Achilles tendons. The left panel of each pair shows a low magnification image of the tendon. A higher magnification of the boxed region is shown in the right panel. (G, H) Control tendons showed the expected columns of tenocytes and very little toluidine blue staining. (I, J) *Ubr5^{mt}* tendons harbour chondrocytes that coincide with regions of toluidine blue staining. (K) Image of control synovium (arrowhead) located underneath the patella (Pa) and patellar tendon (PT) and adjacent to the tibial articular cartilage (AC). (L, M) *Ubr5^{mt}* synovium harbours ectopic tissue. (L) The synovium harbours a bone-like structure (arrowhead). (M) In other regions, the synovium abutting the patella appeared thickened but not ossified showing cartilage harbouring chondrocytes. Pt = patellar tendon; Pa = patella; Mn = meniscus.

3K). Analysis for *lacZ* expression revealed β-gal activity within the upper layer chondrocytes of the femoral and tibial AC (Fig 1Q and 1R) which corresponds with a role for *Ubr5* in this tissue. Thus an overview of the hindlegs at 24 weeks reveals a diverse range of cartilaginous defects which include firstly, metaplastic conversion of connective tissue associated with the knee and ankle and secondly, severe degradation of AC causing exposure of the subchondral bone at the joint surface.

## Ubr5 deficiency results in a postnatal, progressive phenotype

To establish the approximate age at which this striking ECCO phenotype was initially detectable, a timed series of *in vivo* μCT scans on ageing, live animals was followed. *Ubr5^{mt}* animals

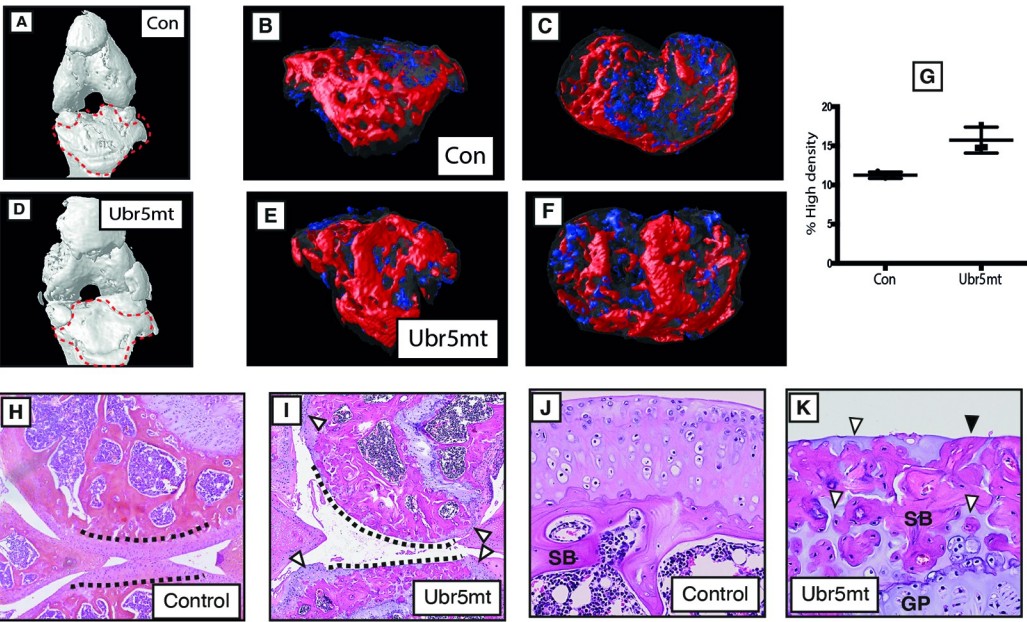

**Fig 3.** *Ubr5^{mt}* **animals show subchondral bone defects and AC cellular and extracellular abnormalities.** (A, D) Surface rendered μCT-based 3D models images of knee joints of *Prx1-Cre* (Control) (A) and *Ubr5^{mt}* (D). Volume rendered 3D models of 26-week-old tibial subchondral bone (outlined in A, D) showing (B, E) ventral, and (C, F) anterior views. High-density volume shown in red and low-density in blue. (G) Graph of percentage of high-density signal volume as a percentage of total subchondral bone volume. s.e.m indicated. n = three biological replicates per genotype. t-test. p = 0.0103. H&E-based histological analysis of (H-K) 26-week-old *Prx1-Cre* (Control) and *Ubr5^{mt}* tissues. (H, I) Low magnification view of the interface at the knee, showing control (dashed lines underscore the AC) and *Ubr5^{mt}* which exhibited extended regions that lacked AC and exposed subchondral bone (dashed lines). Peripheral regions retained some AC (white arrowheads in I). (J) 26-week-old control tibial AC was normal and (K) *Ubr5^{mt}* AC revealed exposed subchondral bone (black arrowheads) and intercalated cartilage (white arrowheads)).

at 3-weeks of age revealed no marked abnormal phenotype in knee or ankle joints (S2A–S2D Fig), suggesting that the ectopic structures did not form during fetal development but rather formed postnatally. Between 6 and 12 weeks of age, the ectopic structures began to emerge (Fig 4A and 4B), initially on the ventral side of the tibia. Dorsally located ectopic signals associated with the AT emerged by 16 weeks of age (Fig 4C) and all ectopic structures were enlarged by 24 weeks of age (Fig 4D). *Ubr5* deficiency, therefore, led to enhanced, progressive chondrogenesis and osteogenesis in the connective tissue of the knee and ankle of the aging *Ubr5^{mt}* mouse.

These metaplastic conversions within the connective tissue supporting the knee and ankle, however, contrasted with the changes demonstrated in the AC which manifested as a degenerative phenotype. To investigate the timing of AC degradation, we examined mice at 3 and 6 weeks. No gross structural disruption of the AC in the *Ubr5^{mt}* animals at 3-weeks of age was detected (Fig 4E and 4F). By 6-weeks of age, *Ubr5^{mt}* articular cartilage exhibited an irregular osteochondral interface (Fig 4G and 4I), clusters of large, hypertrophic-like chondrocytes (Fig 4H and 4J) and a reduction in the number of superficial chondrocytes (Fig 4K). *Ubr5^{mt}* articular cartilage also exhibited multiple tidemarks and regions of strongly eosin positive nuclei indicative of necrosis (Fig 4L and 4M) that were absent in controls. Thus, progressive damage was found in the AC due to the loss of *Ubr5* but in this case, early cellular and extracellular abnormalities occurred prior to AC degradation, which included increased subchondral bone density and exposure of subchondral bone

Despite loss of UBR5 in early limb mesenchyme, these data indicated that the ectopic structures arose postnatally and subsequently progressed with age. To directly address if postnatal UBR5 function was required to suppress ECCO and the degradation of the AC, we utilised a mouse line carrying a tamoxifen-inducible, conditional Cre, *pCAGG-CreERT2* [30]. Control *pCAGG-CreERT2* (*pCAGG-Con*) or *pCAGG-CreERT2;Ubr5^{gt/gt}* (*pCAGG-Ubr5^{mt}*) animals were treated with tamoxifen (administered on two consecutive days) at six weeks of age. Staining for β-gal activity, although more broadly distributed, confirmed tamoxifen-mediated recombination of the *Ubr5^{mt}* gene trap and its associated β-gal expression in tissues that included muscles and tendons (S2E and S2F Fig), and within the midbody ectopia at the AT (S2G Fig). μCT analysis at 8 weeks revealed that tamoxifen-treated control animals exhibited no ectopic signals (Fig 4N), whereas *pCAGG-Ubr5^{mt}* animals exhibited AT -associated ectopic signals (Fig 4O). Scoring (Fig 4P) and heterotopic ossification (HO) volumetric analysis (Fig 4Q) confirmed that only tamoxifen-treated *pCAGG-Ubr5^{mt}* animals exhibited ectopic signals. Comparison of 12 week control to treated *pCAGG-Ubr5^{mt}* (Fig 4R and 4S) knees revealed *Ubr5^{mt}*-associated apical acellular layer (Fig 4S and 4T), damage to the apical surface, multiple tidemarks, reduced superficial zone chondrocytes (Fig 4V) and increased numbers of empty lacunae (Fig 4U and 4W). We concluded that postnatal *Ubr5* function was both necessary and sufficient to maintain AC homeostasis and prevent ECCO.

## Inhibition of *Smo* promotes *Ubr5^{mt}*-associated ECCO and enhances *Ubr5^{mt}*-mediated AC degradation

As UBR5/HYD regulates HH signalling in *Drosophila* [7,8], we next used a genetic approach to address whether aberrant HH signaling contributed to the *Ubr5^{mt}* ECCO and AC phenotypes. The *Smo* gene encodes a core membrane component, regulated by the HH receptor PTCH1, that initiates the downstream signalling cascade leading to GLI-dependent transcription (canonical signalling) or $G_i$ protein-dependent events that are tissue specific (non-canonical signalling). We reasoned that reduction in *Smo* expression levels would sensitize the HH pathway; thus, heterozygosity for a *Smo* loss of function allele (*Smo^{LoF}*) [31] was used in a cross to *Ubr5^{mt}* to create *Prx1-Cre;Ubr5^{gt/gt};Smo^{LoF/+}* animals (*Ubr5^{mt}+Smo^{LoF}*).

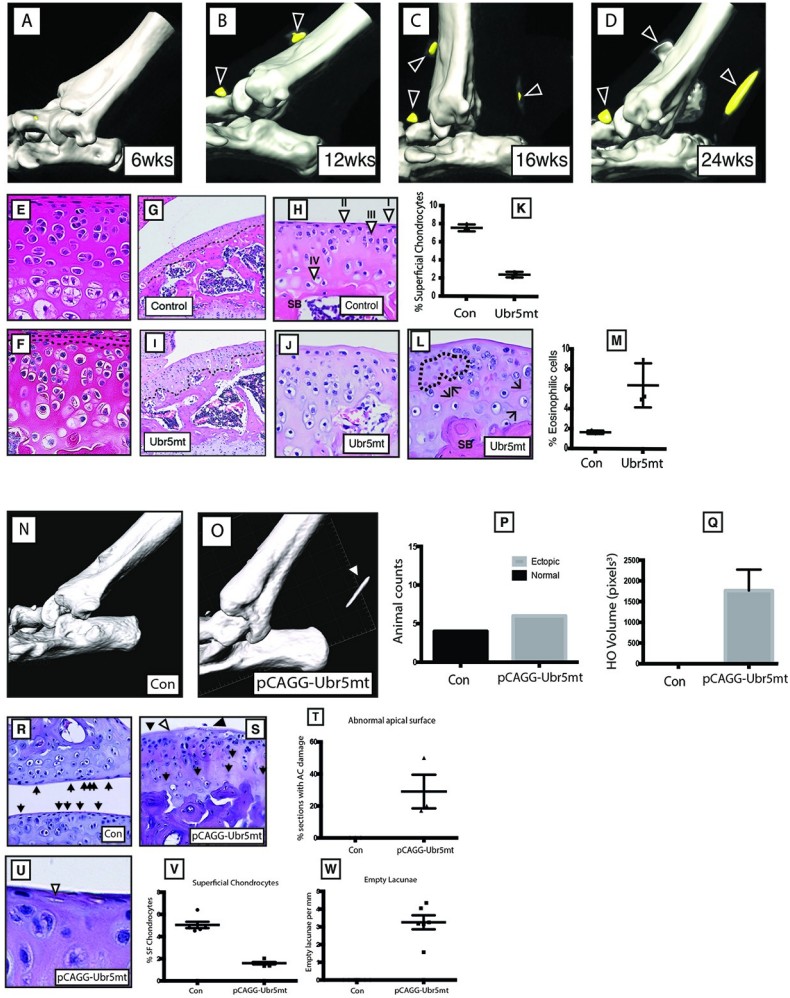

**Fig 4. Ubr5mt mice exhibit degenerative, age related defects.** (A-D) Consecutive uCT scans of live *Ubr5^{mt}* animals at the indicated ages. *Ubr5^{mt}* ankles form (i) ventral ectopic signals (arrowheads) around (B) 12 weeks of age and (ii) dorsal ectopic signals around (C) 16 weeks of age that (D) increased in size over time. Three-week-old knee joints, stained with H&E, of *Prx1-Cre* (control) (E) and *Ubr5^{mt}* (F). Black dashed lines in (F) demarcate the *Ubr5^{mt}* apical acellular region. (G-M) Six-week-old (G,H) *Prx1-Cre* (control) and (I,J,L) *Ubr5^{mt}* proximal tibial AC were analysed. (G) Control AC and at higher magnification (H) revealed the expected chondrocyte profile along the apical to basal axis, namely (I) superficial chondrocytes lining the apical surface; (II) non-hypertrophic rounded/oblong nuclei chondrocytes; (III) larger pre-hypertrophic-like chondrocytes within the central zone; and (IV) large hypertrophic chondrocytes located near the border with the underlying subchondral bone. SB = subchondral bone. (I) *Ubr5^{mt}* tibial AC and at higher magnification (J) revealed abnormal chondrocytes and an acellular apical layer lacking superficial chondrocytes. (K) Graph of the percentage of superficial chondrocytes in six-week-old tibial AC. N = three biological replicates of each genotype. Mean and s.e.m indicated. Chi square test on pooled cell counts. p = <0.0001. (L) *Ubr5mt* tibial AC exhibited clusters of eosin positive chondrocytes (dashed lines) and multiple tidemarks (arrows). (M) Graph of percentage of eosinophilic chondrocytes in six-week-old AC. Chi square test on pooled cell counts. N = three biological replicates of each genotype. p = <0.001. (N-Q) Postnatal *pCAGG-Cre*-mediated recombination of the Ubr5^{gt} construct (O), but not *pCAGG-Cre* expression alone (N), resulted in X-ray dense ectopic signals forming in the AT region. (P) Counts of animals exhibiting ankle-associated ectopic signals, scored for the absence (Normal) or presence of ectopic signals (Ectopic), n = >4 for each genotype. Fisher's exact test, p value = 0.0048. (Q) Volumetric measurement of ectopic signals designated HO (heterotopic ossification) volume from each animal n = 4. Unpaired t test, p = 0.0079. Standard error indicated. Control AC (R) exhibited superficial chondrocytes (arrows). (S) *pCAGG-Ubr5^{mt}* AC exhibited an acellular apical layer (arrowheads), multiple tidemarks (arrows), and surface damage (black arrowhead). (T) Graph of percentage of sections with acellular regions and AC damage. Mean and s.e.m indicated. n = three biological replicates. Three slides analysed from each animal. Average plotted. Fishers exact on pooled section counts. p = 0.0434. (U) *pCAGG-Ubr5^{mt}* AC also exhibited a reduction in superficial chondrocytes and an increase in empty apically-located lacunae (arrowheads). Graph of (V) superficial chondrocytes and (W) empty lacunae expressed as number per mm of AC. n = three biological replicates. Analysis of two sections per animal. Individual slide values plotted. Mean and s.d indicated. Fishers exact test on pooled counts p = <0.0001.

In contrast to our expectations, μCT analysis of 12-week *Ubr5*<sup>mt</sup>+*Smo*<sup>LoF</sup> mice exhibited significantly more severe defects than those of age-matched *Ubr5*<sup>mt</sup> (Fig 5A–5C) and *Smo*<sup>LoF/+</sup> mice (which were indistinguishable from wildtype), with multiple, large ectopic signals apparent around the knee (Fig 5A–5F) and ankle joints (Fig 5H–5N). Volumetric analysis revealed a significant increase in the volume of *Ubr5*<sup>mt</sup>+*Smo*<sup>LoF</sup> femoral-associated ectopic bodies compared to *Ubr5*<sup>mt</sup> alone (Fig 5G) and the ankles harboured a 20-fold increase in the volume of ectopic signals (Fig 5O). In agreement, histological analysis of the *Ubr5*<sup>mt</sup>+*Smo*<sup>LoF</sup> joints revealed an enhanced phenotype to that described in *Ubr5*<sup>mt</sup> (Figs 1 and 2). *Ubr5*<sup>mt</sup>+*Smo*<sup>LoF</sup> synovium harboured large ectopic tissue masses (Fig 6A) with extensive vascularisation (Fig 6B) and chondrocytes lining the surface (Fig 6C) with deeper calcified cartilage and vascularization (Fig 6D). Sagittal sectioning through the ankle revealed large ectopic structures within the superficial digital flexor tendon (Fig 6E), consisting of bone and cartilaginous tissue (Fig 6F and 6H), and at the tendon interface (Fig 6G). Large swathes of chondrocytes were present within the superficial digital flexor and AT that coincided with an absence of tenocytes (Fig 6I and 6J), as previously reported in the *Ubr5*<sup>mt</sup> (Fig 2). In addition, the AC in *Ubr5*<sup>mt</sup>+*Smo*<sup>LoF</sup> knee joints exhibited extensive loss over both tibial and femoral surfaces at this young age (Fig 6M and 6N), while *Ubr5*<sup>mt</sup> knee joints exhibited only tears within the AC (Fig 6K, 6L and quantification in 6O). Importantly, the loss of a single copy of *Smo* alone (*Prx1-Cre;Smo*<sup>LoF/+</sup>) resulted in no structural or AC damage (Fig 6P).

## *Ubr5* suppresses canonical HH signalling and PKA activity

A functional link between UBR5 activity and HH signalling was further examined in 6-week old *Ubr5*<sup>mt</sup> mice. At this age ectopic structures were not detectable (Fig 4), thereby increasing the likelihood of detecting potential causative changes in expression patterns. Immunohistochemistry on *Ubr5*<sup>mt</sup> knee intimal (Fig 7A–7F) and subintimal synovium (Fig 7G–7L) revealed increased *Gli1* expression in comparison to *Prx1-Cre* control animals (Fig 7B, 7E, 7H and 7K; respectively), indicative of increased canonical HH signalling. qRT-PCR analysis also confirmed increased expression of markers of canonical HH signalling in RNA from isolated synovium (*Gli1* and *Ptc1*) (Fig 7M). Additionally, intimal and sub-intimal *Ubr5*<sup>mt</sup> synovium exhibited increased phosphorylated PKA substrate (PPS) staining suggesting decreased $G_i$ proteins activation, characteristic of non-canonical HH signalling (Fig 7C, 7F, 7I and 7L). Consistent with the observations in the synovium, *Ubr5*<sup>mt</sup> AC exhibited markers of increased canonical (Fig 8A–8D) and decreased non-canonical HH signalling (Fig 8E and 8F). Although little change for PTCH1 was detected (Fig 8G) there was significant differences for Gli1 expression and PKA substrate staining (Fig 8H and 8I).

## UBR5<sup>mt</sup> AC and damaged human AC exhibits both aberrant expression of markers of chondrogenesis and HH signalling

As seen in murine *Ubr5*<sup>mt</sup> AC, osteoarthritic AC from patients also exhibits markers of increased canonical HH signalling [32]. We next addressed (i) UBR5 expression and (ii) markers of decreased non-canonical HH signalling (PPS) in human AC. Graded samples from (OA) patients (S3A–S3C Fig) undergoing total joint replacement were assessed for UBR5 expression (S3E, S3G and S3I Fig) and PKA activity (PPS in S3D, S3F and S3H Fig). As in the murine model, PPS IHC staining increased (S3J Fig), and hUBR5 staining decreased (S3K Fig) with decreasing AC health. Observations of changes in markers consistent with increased canonical and decreased non-canonical HH signalling in *Ubr5*<sup>mt</sup> synovium and AC were echoed in human OA samples.

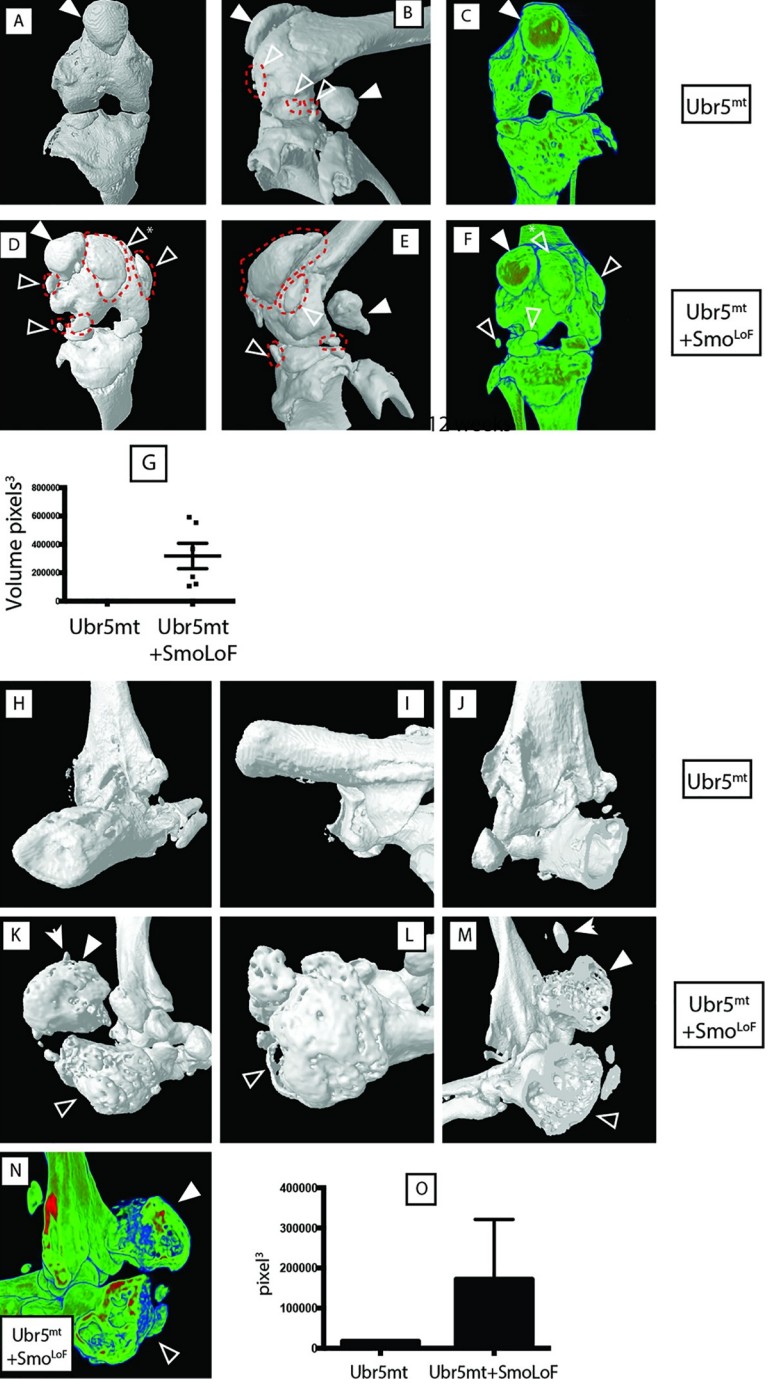

**Fig 5. *Smo<sup>LoF</sup>* enhances the *Ubr5<sup>mt</sup>* ECCO phenotype.** Analysis of 12-week-old knee (A-G) and ankle (H-O) joints by μCT-based 3D models. (A-C) *Ubr5<sup>mt</sup>* and (D-F) *Ubr5<sup>mt</sup>*+*Smo<sup>LoF</sup>* knee joints revealed ectopic structures marked by red dashed lines and open arrowheads. Sesamoid bones indicated by closed arrowhead. Asterisk marks an ectopic structure displacing the patella. Images (A, B, D, E) are surface rendered 3D models, while (C, F) are optical cross sections through the volume-rendered model revealing the internal structure and X-ray densities of the (open arrowhead) ectopic sesamoid and (closed arrowhead) abnormal patella structures. (G) Volumetric analysis of ectopic structures revealed *Ubr5<sup>mt</sup>*+*Smo<sup>LoF</sup>* exhibited a dramatic increase in total ectopic volume over *Ubr5<sup>mt</sup>* alone. Mean and s.e.m indicated. n = six knees from three animals for each genotype. t-test. p = 0.0002. (H-J) *Ubr5<sup>mt</sup>* ankle joints exhibited a few small ectopic signals. (K-N) *Ubr5<sup>mt</sup>*+*Smo<sup>LoF</sup>* ankles joints exhibited large (closed arrowhead) and small (arrow) ectopic signals in addition to an abnormal and enlarged calcaneus (open arrowhead). (N) Optical cross sections through volume-rendered model revealed the internal structure and X-ray densities of the (open arrowhead)

calcaneus and (closed arrowhead) ectopic structure. (O) Volumetric quantification of ectopic structures in the indicated genotypes. n = five animals per genotype. t test, p value = 0.0293. Mean and s.e.m. indicated.

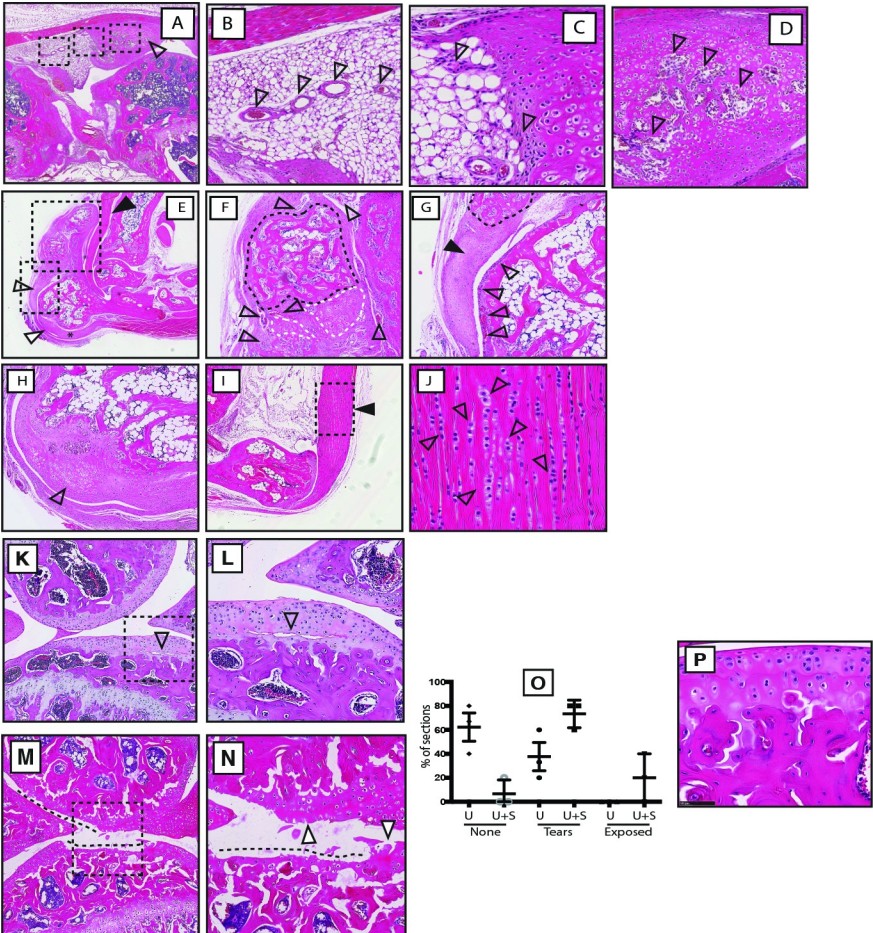

**Fig 6.** $Smo^{LoF}$ **enhances the** $Ubr5^{mt}$ **AC phenotype.** (A) $Ubr5^{mt}+Smo^{LoF}$ synovium exhibited large ectopic tissue deep to the patella and adjacent to the femur (open arrowhead). Three black dashed boxes, from left to right, are enlarged in (B), (C) and (D), respectively. (B) Sub-intimal synovial layer abutting the ectopic tissue was highly vascularized (arrowheads). (C) The region interfacing with the ectopic tissue harboured plump spindle-like cells and chondroid-like cells (arrowheads). (D) The core of the ectopic tissue resembled calcified cartilage undergoing endochondral ossification and harboured vascularized cavities (arrowheads). (E) $Ubr5^{mt}+Smo^{LoF}$ ankle exhibited a large ectopic structure (black arrowhead) and abnormal superficial digital flexor tendon (open arrowhead) and calcaneus (asterisk). The upper and lower dashed boxed region are enlarged in (F) and (G), respectively. (F) The ectopic mass contained bone-like (black dashed lines) and cartilaginous tissues (white dashed lines) surrounded by extensively vascularised synovium (open arrowheads). (G) The presumed superficial digital flexor tendon attached to the ectopic bone (black arrowhead), harboured chondrocytes and resembled cartilage (encircled by black dash line). The adjacent periosteum of the calcaneus was highly vascularised (open arrowheads). (H) Cartilaginous thickening of the outer calcaneus. (I) The AT was thickened. The dashed box enlarged in (J) shows columns of chondrocytes (open arrowheads). (K-N) Analysis of 12-week-old knee joints of $Ubr5^{mt}$ and $Ubr5^{mt}+Smo^{Lof}$ by H&E stained histological sections of the lateral condyles. (K, L) $Ubr5^{mt}$ exhibited tears in the AC (open arrowhead) and (M, N) $Ubr5^{mt}+Smo^{LoF}$ exhibited extensive loss of AC (dashed lines) and damaged apical surfaces (arrowhead). Dashed boxes in (K) and (M) indicate the enlarged regions in (L) and (N), respectively. (O) Percentage of sections bearing no damage ('None'), tears ('Tears') or exposed calcified cartilage (Exposed), revealed a significance difference between the genotypes. Mean and s.e.m indicated. Chi-square test on pooled slide counts. p = 0.0027. (P') $Smo^{LoF}$ AC showed no signs of AC damage.

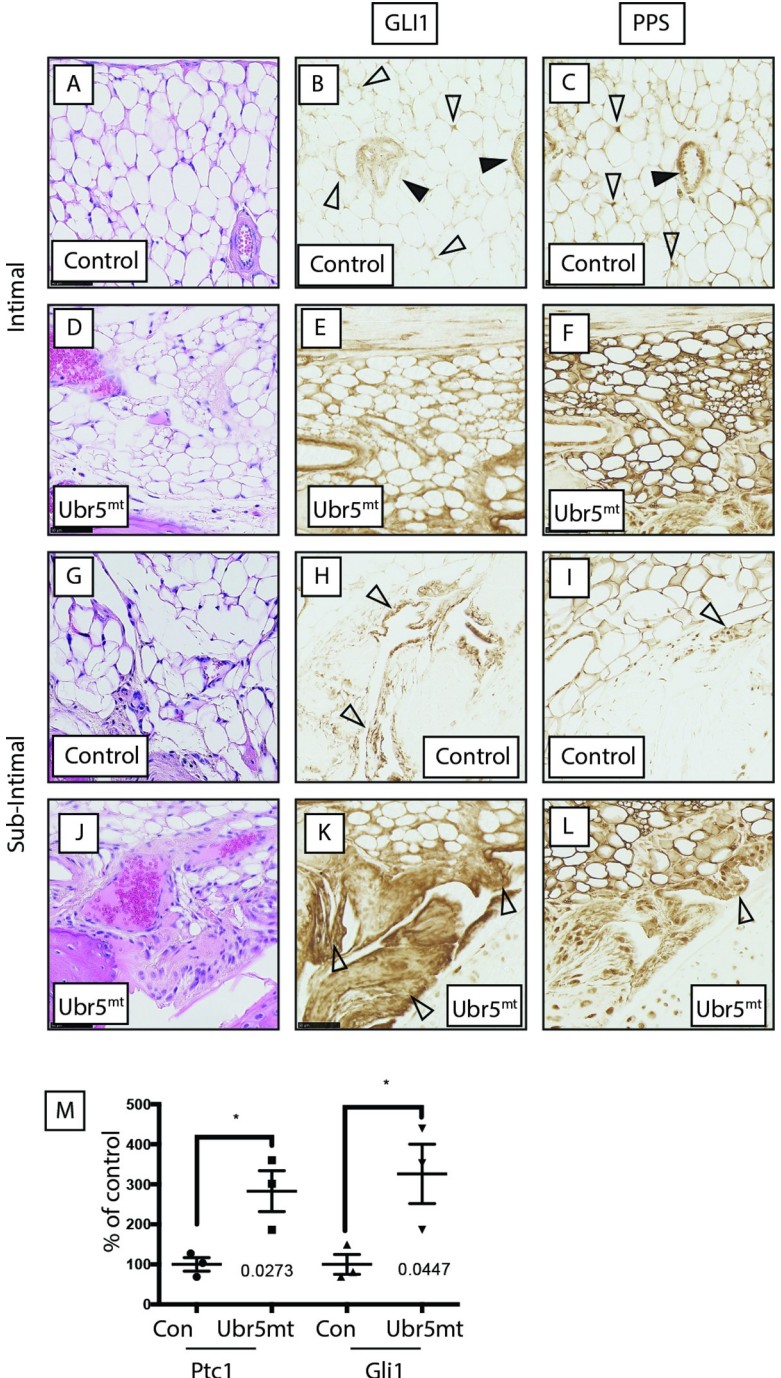

**Fig 7. *Ubr5^mt* synovium exhibits markers of increased canonical and decreased non-canonical HH signalling.**
Immunohistochemical localization of markers of canonical and non-canonical HH signalling (A-L) six-week-old
sagittal sections of *Prx1-Cre* control (Con) and *Ubr5^mt* animals. In general, control (A-C) intimal and (G-I) subintimal
layers exhibited weaker GLI1 and PPS staining than in comparable *Ubr5^mt* sections (D-F and J-L, respectively). (B)
GLI1 staining in the control intimal layer was located to the vasculature (closed arrowheads) and some adipocytes
(open arrowheads). (C) PPS staining was in the vasculature (closed arrowheads) and within adipocytes (open
arrowheads). (E) GLI1 and (F) PPS staining were throughout the subintimal layer. (H) GLI1 staining of control
synoviocytes within the subintimal layer (arrowhead). (I) PPS staining in sub-intimal layer synoviocytes (arrowhead).
(K) GLI1 and (L) PPS staining were strongly expressed within hyperplastic and thickened synovial sub-intimal layer.
Synoviocytes exhibited robust nuclear and cytoplasmic staining for GLI1 (K) and PPS (L). (M) qRT-PCR on synovium
RNA for expression of canonical HH pathway expression markers *Ptch1* and *Gli1*. Graph indicates mean and s.e.m.
n = three animals. t-test. *Ptch1* p = 0.0273 and *Gli1* p = 0.0477.

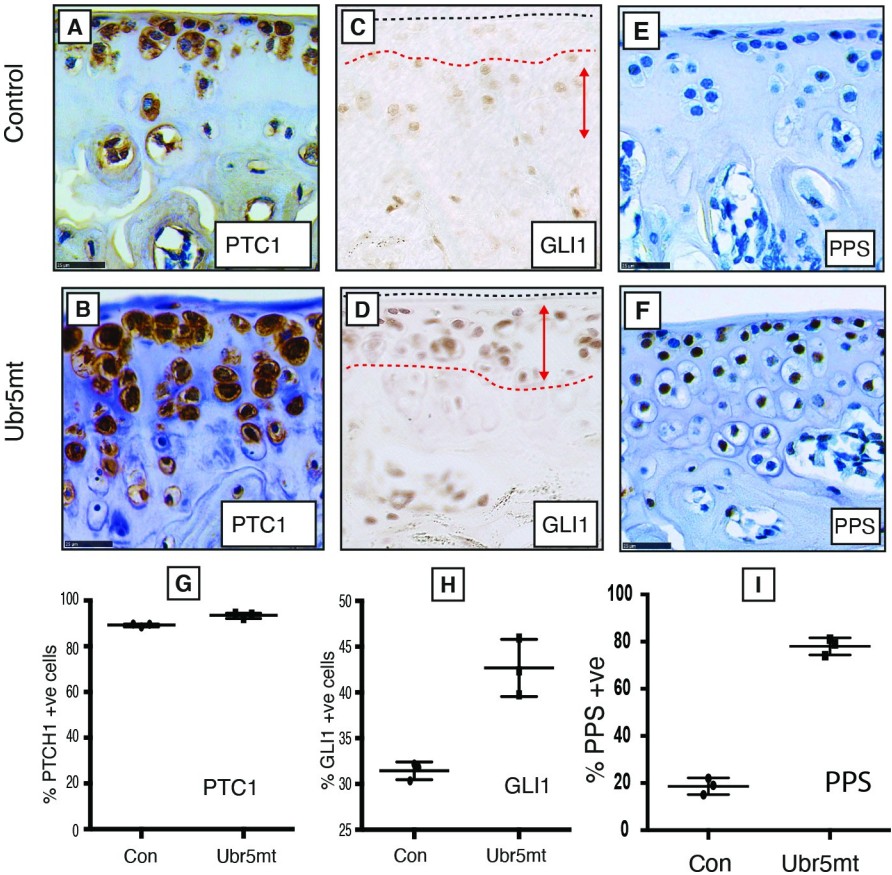

**Fig 8. Impaired Ubr5 function results in increased canonical and decreased non-canonical HH signalling.**
Immunohistochemical analysis of six-week-old control and *Ubr5^{mt}* tibial AC examined for markers of canonical HH pathway activity. Relative to (A, C) control, (B, D) *Ubr5^{mt}* AC displayed increased staining intensities for PTCH1 (A, B) and GLI1 (C, D) with GLI1 exhibiting expanded expression domains (D, double-headed arrows). (E, F) Staining for PKA phosphorylated substrates (PPS) revealed (Q) *Ubr5^{mt}* AC exhibited increased numbers of robust staining cells. The number of expressing cells is quantified in (G-I). Quantification confirmed *Ubr5^{mt}* AC to harbour increased numbers of positive cells for all antigens except PTCH1. Graphs represent the percentage of positive cells, regardless of staining intensity, with the mean and s.e.m indicated. n = 3 biological replicates. Chi-square test on pooled cells count data. p = <0.0008, except PTCH1 which was not significant.

To further delineate whether mammalian *Ubr5* could influence markers of canonical and non-canonical HH signalling, murine NIH3T3 cells were engineered to either exhibit increased (cDNA overexpression) or decreased (shRNA knock-down) *Ubr5* expression. Cells were then transfected with constructs encoding (i) *Shh*, (ii) constitutively active *Smo* mutant (*Smo-M2*) or (iii) *Gli1*. Canonical pathway activity was measured using a *Gli*-responsive luciferase reporter assay. While perturbation of *Ubr5* expression had no effect on Shh- or Smo-M2-mediated signalling (Fig 9A and 9B), *Ubr5* overexpression caused a significant reduction (Fig 9A, P<0.001), and *Ubr5* shRNA-mediated knockdown caused a significant increase (Fig 9B, P<0.05), in Gli1-mediated luciferase activity. However, *Ubr5*-overexpression did not perturb the expression level of endogenous or exogenous GLI1 protein (Fig 9C), excluding a role for UBR5-mediated degradation. Therefore, UBR5 appeared to only suppress canonical HH signalling associated with overexpression of GLI1.

We then addressed whether loss of *Ubr5* function would also affect cAMP production as a readout of $G_i$ protein activity, an indirect marker of non-canonical HH signalling. *Ubr5* shRNA cells showed an ~2-fold increase in maximal cAMP production in response to

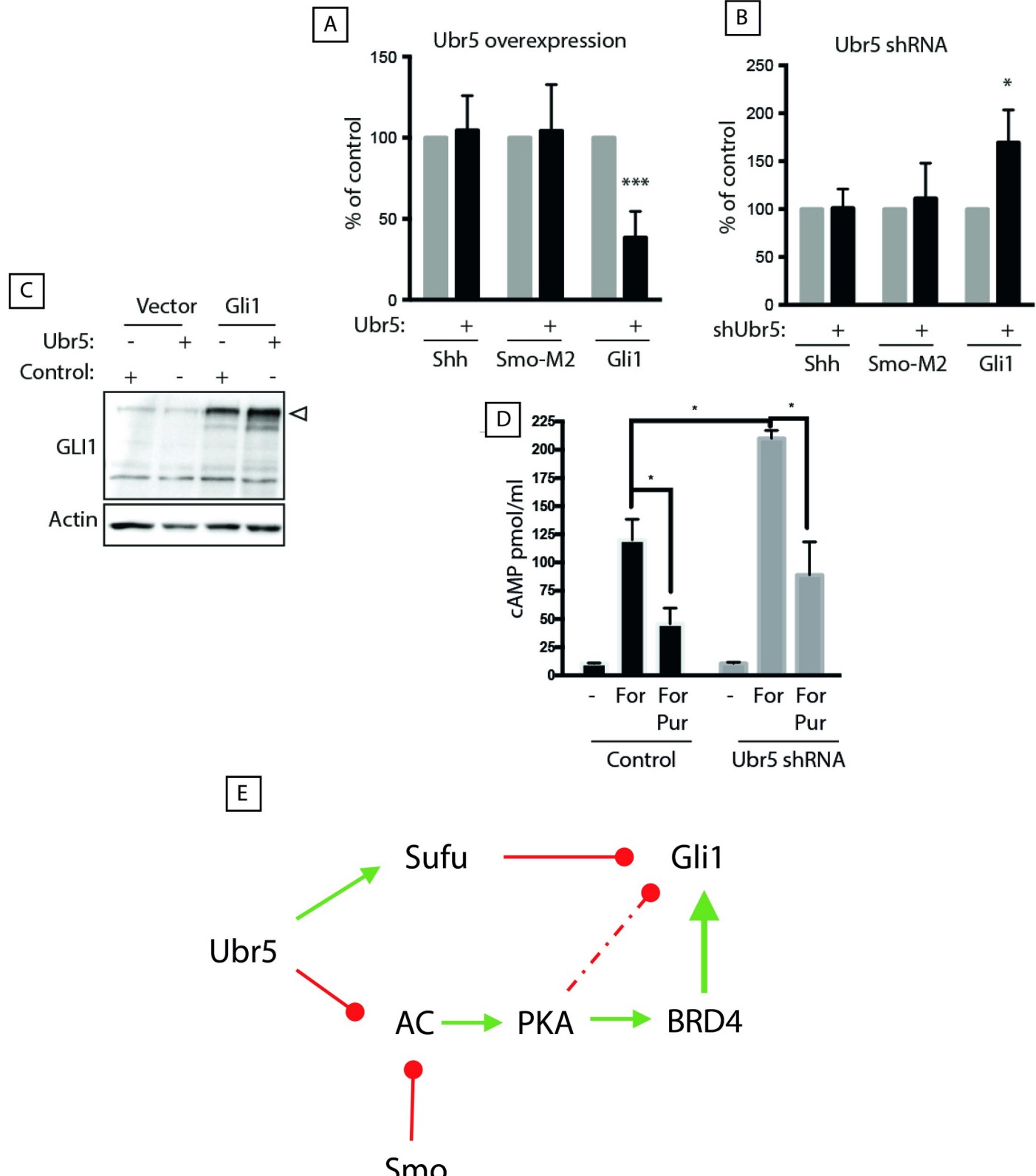

**Fig 9. Ubr5 functions as a negative regulator of HH signalling ex vivo.** Analysis of HH pathway activity in murine NIH3T3 cells in response to modulation of Ubr5 expression. (A) Cells were transfected with empty pN21 vector (grey bars) or pN21-*Ubr5* (black bars) together with plasmids encoding *Shh*, *Smo-M2* or *Gli1*

and 8xGLI-*Firefly* and pTK-*Renilla* luciferase reporters in growth medium (DMEM with 10% FBS). After 24 h, serum was reduced to 0.5% and the Firefly/Renilla luciferase activity was measured 48 h later. Bars represent mean +/- s.e.m. of n = 3 independent experiments. (B) A similar GLI-luciferase assay was carried out in NIH3T3 cells stably expressing *Ubr5* shRNA (black bars) or *scrambled* shRNA (grey bars). Bars represent mean +/- s.e.m. of n = 3 independent experiments. (C) NIH3T3 cells were co-transfected with pN21-*Ubr5* (Ubr5) or empty vector (Control) and *Gli1-myc* or empty pcDNA3.1, followed by Western blot analysis of Gli1 expression (arrowhead) using β-actin as loading control. (D) Stable knockdown of *Ubr5* impaired readouts of non-canonical HH signalling. Production of cAMP by control *scrambled* shRNA (black bars) or *Ubr5* shRNA stable cells (grey bars) following acute treatment with the adenylate cyclase activator forskolin (For) or forskolin plus the SMO agonist purmorphamine (For/Pur), compared to DMSO vehicle as control (-). Forskolin-stimulated cAMP production in *Ubr5* shRNA cells was significantly elevated compared to control cells (p = 0.0368; *t*-test). Purmorphamine suppressed forskolin-mediated cAMP production in both scramble control (p = 0.0318; *t*-test) and *Ubr5* (p = 0.0160; *t*-test) shRNA cell lines. Graphs indicate mean and s.e.m.; n = 4 independent experiments. (E) Proposed model of UBR5 function in HH signalling: UBR5 negatively regulates canonical HH signalling downstream of SMO, hypothetically through facilitating the function of the HH negative regulator Sufu, despite simultaneously inhibiting adenylate cyclase (AC). In this context, loss of Ubr5 could increase Gli1 expression by two means: 1) impairment of Sufu negative regulation and 2) stimulation of Gli1 transcriptional activity by increasing PKA-dependent phosphorylation of BRD4. The convergence of Ubr5 and SMO to suppress adenylate cyclase activity could explain the phenotypic enhancement observed in compound mice with loss of function of Ubr5 and Smo. Green and red arrows indicate established modes of activation and repression, respectively.

forskolin, an adenylate cyclase agonist (Fig 9D) [33]. Moreover, simultaneous addition of forskolin and purmorphamine, a SMO agonist, lowered maximal cAMP generation, but its effect was suppressed by *Ubr5* shRNA (Fig 9D). Together, the *in vitro* findings suggest that Ubr5 loss results in reduced stimulation of $G_i$ proteins by Smo, leading to increased cAMP/PKA activity levels. Overall, these data supported our *in vivo* observations that *Ubr5* normally acts to suppress GLI1 activity while promoting PKA activity.

## Discussion

### *Ubr5* mutation causes musculoskeletal tissue defects

We report a role for mammalian *Ubr5* in adult skeletal homeostasis that impacts upon and genetically interacts with, components of the HH signalling pathway. These findings add to the emerging importance of the N-end rule ligases in regulating important signaling and cellular processes in human, and animal health and disease [34,35]. Loss of the *Ubr5* gene in early limb mesenchyme resulted in postnatal defects in and around joints within the fore and hindlimb. The mutant phenotype displayed a degree of variability between individual mice; however, the abnormalities were pervasive in the mutants and were never observed in wildtype or control mice. The defects that were consistently observed included ectopic bone and cartilage formation, and articular cartilage degradation. The spectrum of defects observed are summarized in S4 Fig.

Our data indicates metaplastic production of chondrocytes and/or ectopic endochondral ossification as a major component of *Ubr5mt*-associated ECCO. Comparison of the *Ubr5mt*-associated ECCO phenotype with that of human inherited HO diseases reveals some similarities and differences. Within the ECCO-prone tissues there were distinct tissue-specific responses; for example, the knee-associated synovium underwent ectopic chondrogenesis, calcification and ossification to produce bone, whereas the Achilles tendon only underwent ectopic chondrogenesis and calcification. The abnormalities of the knee-associated synovium which display heterotopic chondrogenesis are reminiscent of human benign bone tumours called osteochondromas [36], whereas the heterotopic tissue calcification without ossification seen in the AT resembles a form of calcific tendinopathy [37]. The mouse *Ubr5* mutation, thus, provides a genetic model for the generation of these bone abnormalities and suggests that the processes of chondrogenesis, tissue calcification and ossification represent discrete, albeit interrelated, steps that when deregulated can individually, or collectively, contribute to distinct tissue pathologies.

Our findings also demonstrated an important role for Ubr5 in regulating AC homeostasis, where its loss led to dramatic cellular, extracellular and structural defects. The observed defects

in HH signalling could have been causative in nature as HH signalling is intimately linked to both stem cell [22] and chondrocyte biology [10]. One of the more distinctive *Ubr5^mt* AC defects was the tearing along the tidemark between non-calcified and calcified cartilage. This focal failure suggested the interface was prone to transverse shear forces and 'slipping' of one layer (i.e., non-calcified cartilage) relative to the other (i.e., calcified cartilage). Interestingly, this mode of AC shedding and the associated regions of necrosis mirrored defects observed in mammalian osteochondrosis [38,39].

Recently, increased *Ubr5* expression was correlated with muscle growth, hypertrophy [40] and recovery from injury [41] suggesting a role in muscle maintenance. These reports suggest that *Ubr5* may have a broad role in the musculoskeletal system of the limbs affecting homeostasis not only in the cartilaginous connective tissue but also in the associated musculature.

## UBR5 influences markers of canonical and non-canonical HH signalling

Based on the current dogma, we hypothesized that the *Ubr5^mt*-associated ECCO was caused by increased HH signalling. In contrast, the introduction of *Smo^LoF* heterozygosity into a *Ubr5^mt* background both (i) exacerbated *Ubr5^mt*-associated defects as well as elicited novel defects not observed by loss of *Ubr5* function alone (e.g., ECCO of the calcaneal periosteum and the superficial digital flexor tendons and increased volume and altered shape of normotopic sesamoid bones). This combined ability to influence both normotopic and heterotopic bones (S4 Fig), highlights the importance of *UBR5* in normal and pathological skeletal tissue homeostasis. Furthermore, our genetic analysis exposed a pro-homeostatic function for SMO– and by extension HH signaling–in suppressing *Ubr5^mt* ECCO.

*In vivo* and *in vitro* observations identified a loss of *Ubr5* associated with predictors of increased (GLI1 activity) and decreased (PKA activity) canonical HH signalling. Based on the current dogma, it is difficult to reconcile increased GLI activity in the context of increased PKA activity, given that PKA phosphorylates other GLI family members, GLI2 and GLI3, targeting them for processing into transcriptional repressors [14,42]. However, the evolving breadth of the HH pathway (Fig 9E) provides potential mechanistic explanations for this apparently paradoxical observation.

Recent evidence expanded the role of PKA to promote canonical HH signalling by promoting BRD4-mediated stimulation of GLIs transcriptional activity (Fig 9E) [43–45]. Interestingly, HO-associated with increased HH signalling was suppressed by the BRD4 inhibitor JQ1 [46], which clearly demonstrated a role for a cAMP-PKA-BRD4-GLI1 axis in skeletal tissue homeostasis. A non-canonical role of SMO as a G protein-coupled receptor (14, 15) provides a mechanism to control PKA activity. Upon stimulation, SMO activates heterotrimeric $G_i$ proteins, which, upon dissociation, inhibit adenylate cyclase through the $G\alpha$ subunit to reduce cAMP production and PKA activation [15,47,48]. Therefore, SMO inhibition can lead to increased cAMP-mediated PKA activity accounting for SMO modification of the *Ubr5^mt* phenotype, as impairment of either UBR5 or SMO leads to increased cAMP-mediated PKA activity–with their combined impairment leading to either additive or synergistic effects. Interestingly, our preliminary research (personal communication NDGR) supports a role for UBR5 in regulating readouts of non-canonical HH signalling other than PKA (i.e.; RhoA) [16]. Although our data reveal a genetic interaction between UBR5 and an essential component of the HH signalling pathway, we cannot fully establish the underlying mechanism(s) driving *Ubr5^mt*-associated ECCO. Future work will require developing the tools to differentiate between causative individual, or combined, contributions of aberrant canonical or non-canonical HH signalling. The addition of *Smo^LoF* into a *Ubr5^mt* background would have exacerbated a pre-existing imbalance between the pathway outputs to drive ECCO.

The importance of balanced canonical and non-canonical HH signalling was recently demonstrated in osteogenesis [49]. Loss of the cilia regulatory protein IFT80 resulted in impaired osteoblast differentiation and coincided with (i) decreased expression of canonical target genes and (ii) increased non-canonical activity. The authors proposed that the non-canonical HH pathway prevented, and the canonical pathway promoted, formation of osteoblasts. Due to the emerging importance of non-canonical HH signalling [12], we also propose that the combined effects on canonical and non-canonical HH signalling contributed to the observed loss of tissue homeostasis in *Ubr5$^{mt}$* animals. Overall, our detection of *Ubr5$^{mt}$*-associated increased canonical (GLI1 activity) and indications of decreased non-canonical HH signalling (cAMP-PKA) are in general agreement with a reported pro-osteogenic environment conducive to HO [49]. UBR5 may therefore join IFT80 [49] and DYRK1B [50] as differential regulators of canonical and non-canonical HH signalling. Our future work will involve establishing which of the various non-canonical, SMO's GPCR-associated downstream effectors (e.g., PKA, RHOA, RAC1, PI3K etc.) [51,52] drive ECCO.

In summary, we reveal a previously unknown role for *Ubr5* in influencing HH signalling, tissue homeostasis and preventing spontaneous ECCO. A role for UBR5 in regulating HH signalling and tissue homeostasis supports the classification of human *UBR5* as a Tier 1 human cancer susceptibility gene (Sanger Cancer Gene Consensus). We believe the *Ubr5$^{mt}$* mouse model could assist in uncovering mechanisms that lead to disorders including characterisation of early pathological events and elucidation of pro-homeostatic mechanisms capable of promoting general bone health. In the future, manipulation of human *UBR5* and *SMO* function could potentially provide a means of preventing pathological, and promoting beneficial, chondrogenesis and ossification in both the clinic and in biomedical engineering applications.

## Materials and methods

### Ethics statement

**Human material.** Human AC was obtained from knee joint arthroplasty specimens with ethical approval from the Lothian Research Ethics Committee. Written formal consent was obtained.

**Murine studies.** All animal experiments were reviewed and approved by the University of Edinburgh Animal Welfare and Ethics Committee and were conducted with appropriate licensing under Animals (Scientific Procedures) Act 1986 (license number PPL 60/4424).

*Prx1-Cre;Ubr5$^{gt/gt}$* experimental animals (referred to as *Ubr5$^{mt}$*) and their respective littermate controls were generated and all experiments were conducted in accordance with the ARRIVE guidelines. Animals were routinely weighed and there were no significant differences between experimental and control animals. Males animals were used for analysis unless otherwise stated. Tamoxifen (0.1mg/kg body weight) in corn oil, or vehicle only, were administered i.p to six-week-old animals on two consecutive days. For X-gal staining, embryos and postnatal hind limbs were dissected, fixed in 4% formaldehyde (from paraformaldehyde [PFA]) at 4˚C, washed and stained in X-Gal stain solution (XRB supplemented with 1mg/ml X-Gal) overnight [20].

### Histology

Hindlimbs were fixed in 4% formaldehyde (fromPFA)) for 72hrs at 4˚C before being decalcified 0.5M ethylenediaminetetraacetic acid (EDTA) pH7.4 at 4˚C. Samples were embedded in paraffin wax blocks and 5μm sagittal sections cut. For cryotome sectioning, samples were equilibrated in a 30% sucrose/phosphate buffered saline (PBS) solution at 4˚C and then embedded in OCT compound (Fisher Scientific, Loughborough, UK) before 10μm sagittal

sections were cut. For human material, 8x3mm blocks of AC were cut from femoral tibial condyles and fixed in neutral buffered formalin and then paraffin wax embedded. Histological staining with Von Kossa (Abcam, Cambridge, UK), toluidine blue (Sigma) and haematoxylin and eosin (Sigma) were carried out according to standard procedures. All histological scoring was carried out on the lateral tibial condyle with AC damage determined by a binary scoring system, of 'normal' or 'damaged'. At least three slides separated by 25μm were analysed for each limb. For cell and immunohistochemical scoring, cell-types or positive staining cells were expressed as a percentage of the total chondrocyte count. The number of empty lacunae were expressed per mm of AC analysed.

## Immunohistochemistry

*Primary antibodies*: rabbit anti-IHH (1:200, Millipore, Billerica, US); goat anti-PTCH1 (1:50, Santa Cruz, Dallas, US); rabbit anti-GLI1 (1:50, Cell Signalling); rabbit anti-SOX9 (1:50 Santa Cruz); rabbit anti-RUNX2 (1:250, Sigma); PKA phosphorylated substrates (1:150, Cell Signalling); rabbit anti-EDD1 (HsUBR5) (1:100, Bethyl Labs, Montgomery, US). Biotinylated secondary antibodies: goat anti-rabbit and horse anti-goat (1:200, Vector Labs).

Paraffin sections were de-waxed, blocked for endogenous peroxidase and underwent antigen retrieval in 10mM sodium citrate pH6 at 80˚C for 30–60 minutes. Slides were blocked with serum-free pan-species block (DAKO, Glostrup, Denmark), incubated with primary antibodies overnight at 4˚C, and incubated with biotinylated secondary antibodies for 45mins at room temperature. Sections underwent streptavidin-mediated signal amplification (ELITE ABC, Vectorlabs, Burlingame, US) prior to incubation with peroxidase substrate kit DAB (Vectorlabs).

## μCT monitoring

For longitudinal studies, eight-week-old mice were anesthetized with isoflurane prior to in-vivo μCT scanning using a Skyscan 1076 (Bruker, USA, MA) at a resolution of 18μm isotropic voxel size. Procedures were repeated every four weeks and carried out under local ethical approval (ERF WGH-14-74). Fixed limbs were imaged at 18μm resolution using a Skyscan 1076 (Brucker, USA, MA) at 65kV, 110uA, 0.5mm Al filter, 700ms exposure, 0.6-degree rotation step, 180-degree rotation, 2 frame averaging.

## μCT image processing

Raw μCT image stacks was reconstructed using GPU-based NRecon (Bruker, USA, MA) using identical parameters within each study type (10% beam hardening and image conversion range of 0.003–0.1125. Reconstructed image stacks were imported into CTAn (Bruker, USA, MA) for selecting regions of interest and acquiring 2D density maps, volumetric quantification of ectopic structures and generation of surface rendered 3D models. Surface rendered 3D were visualized in CTVol (Bruker, USA, MA). 3D volume rendered models were generated in CTVox using a colour-coded transfer functions to identify low, medium and high densities. Longitudinal studies used IMARIS (Bitplane, UK) to create, view and perform volumetric quantification of 3D models.

## RNA extraction and q-RT-PCR analysis

Individual joint components were micro-dissected and stored in liquid nitrogen. RNA was extracted using Trizol reagent (Life Technologies), according to manufacturer's instructions. RNA was reverse-transcribed using QuantiTect Reverse Transcription Kit (Qiagen). The

qRT-PCR was performed using LightCycler 480 SYBR Green I Master (Roche, Germany) and target gene expression normalized to *Rpl5* and analysed using the ΔΔCT method [53].

## Plasmid constructs

The *Shh* and *SmoM2* (W593L) expression vectors were provided by P. Beachy (Stanford University, USA, CA). *mGli1* expression and the reporter vectors *8xGBS-luc* were a gift from H. Sasaki (Osaka University, Japan). *pCMV-dR8.2 dvpr* (8455) and *pCMV-VSV-G* (8454) were generated in the Weiner lab and obtained from Addgene (USA). *pRL-TK* was obtained from Promega (USA) and *pcDNA3.1+* was purchased from Invitrogen (USA). Recombinant SHH ligand was synthesized and purified as described previously [54].

The complete *Ubr5* cDNA was synthesised from murine embryonic stem cells total RNA [9] and cloned into a modified pcDNA5/FRT vector (Life Technologies) containing an amino-terminal 2×HA/2×Strep. NIH3T3 cells (American Type Culture Collection, USA) were seeded at a density of 100,000/ml and transfected after 24hr with *pcDNA3.1* alone, *UBR5* and *pcDNA3.1*, *Gli1* and *pcDNA3.1*, or Gli1 and *Ubr5* using FuGENE6 (Roche). After 48hrs, the medium was replaced by DMEM/0.5% FCS, and cells were lysed 24 hrs later in Laemmli buffer. Whole cell lysate was separated on a 6% SDS-PAGE and transferred onto nitrocellulose membranes. Membranes were blocked in 5% non-fat milk, incubated with primary antibodies overnight at 4˚C at 1:1,000 dilution for GLI1 (Cell Signalling) or 1:10,000 dilution for β-actin (Sigma). Secondary HRP-conjugated-anti-mouse antibody was applied at a 1:2,000 dilution for 1hr at room temperature. The membranes were developed using the Clarity western ECL substrate (BioRad, USA, CA).

## Retrovirus production and stable *Ubr5* silencing

Previously validated shRNA-encoding oligos targeting murine *Ubr5* and or a scrambled sequence were cloned into *pLKO.1-puro* (Sigma). sh*UBR5* and *shScrambled-pLKO.1-puro* were co-transfected with *pCMV-VSV-G* and *pCMV-dR8.2 dvpr* plasmids into HEK 293T cells using TransIT293 reagent (Mirus Bio LLC, USA). To generate stable silenced sh*UBR5* cells, NIH3T3 cells were seeded at 120,000 cells/ml and infected with 0.5 ml sh*Scramble* or sh*UBR5* retroviral supernatant in the presence of 8 mg/ml polybrene (Sigma). The media was changed after 24 hrs and cells were selected with 2 mg/ml puromycin 48 hrs post-infection.

## *Gli*-luciferase assay

NIH3T3 cells were seeded, and after reaching 70% confluence transfected with *pcDNA3.1*, *Shh*, *SmoM2*, or *Gli1* together with *Gli*-luciferase and *Renilla luciferase* reporter plasmids with or without *pcDNA5-HA-Strep-Ubr5*, using FuGENE 6 transfection reagent (Roche) according to the manufacturer's protocol. For *Ubr5* knockdown studies, stable *shScramble* and *shUbr5* NIH3T3 cells were transfected with *pcDNA3.1*, *Shh*, *SmoM2*, or *Gli1*, together with *Gli*-luciferase and Renilla luciferase reporter plasmids. In both cases, after the cells reached 100% confluency, the medium was replaced with DMEM/0.5% FCS. After 24 hrs, Firefly and Renilla luciferase activities were determined with the Dual Luciferase Reporter Assay System (Promega).

## cAMP assay

Control (scrambled) or knock down (*Ubr5* shRNA) NIH3T3 cells were seeded at 130,000 cells/ml, serum starved overnight, and stimulated with 10μM forskolin (FORSK) for 5min. Cells were pre-incubated with 5μM purmorphamine for 10min before addition of FORSK. Cells

were processed according to Parameter cAMP Enzyme Immune Assay (R&D Systems) instructions.

## Statistical analysis

Data analysis and statistics was performed using PRISM software (GraphPad, La Jolla, US). Count data was analysed using a contingency table and either two-sided Chi square or Fisher's exact tests according to count size. Continuous data was analysed using unpaired, two-tailed Students t-tests. The level of significance for all tests was set at p = <0.05.

## Supporting information

**S1 Fig. *Ubr5^{mt}* mice exhibit gait abnormalities and ectopic X-ray-dense signals and analysis of subchondral bone.** 24-week-old control or *Ubr5^{mt}* mice were assessed for (A-C) behavioral analysis. (A,B) Mice were videoed while walking along a boxed runway and their static positioning recorded as either 'sprung' (with their posterior not in contact with the floor), or 'squat' (with their posterior resting on the floor). (C) Graph showing counts of animal behavior. n = six of each gender for controls and eight of each gender for the *Ubr5^{mt}* genotype. Fisher's exact test, p value = <0.0001. (D) Control ankles and (E) *Ubr5^{mt}* ankles which exhibited ventrally- and dorsally located isolated signals. Dashed box region enlarged in indicated panel. (F) Control knee joints exhibited the fabella, a dorsally-located sesamoid bone (open arrowhead). (G) *Ubr5^{mt}* knee joints exhibited a misshapen fabella (open arrowhead), with the dashed boxed region being enlarged in the indicated panel to the right). n = eight males and eight females. (H-I) show the volume rendered 3D models of 26-week-old tibial subchondral bone. Total subchondral bone volume used in the analysis is shown in grey (H) and the high-density signal in red (I) and the two merged (K).
(TIF)

**S2 Fig. Ectopic structures are not detected in three-week-old control or *Ubr5^{mt}* ankle or knee joints and require postnatal expression of *Ubr5*.** (A-D) Different views of surface rendered 3D models of three-week-old control and *Ubr5^{mt}* (A,B, respectively) knee and (C,D, respectively) ankle joints. (A-D) From left to right panels: ventral, dorsal, medial and lateral views. Both control and *Ubr5^{mt}* joints exhibit either a normal array of sesamoid bones, developing epiphysis and calcifying menisci. (E-G) Analysis of 18-week-old tamoxifen-treated *pCAGG-Cre* control and pCAGG-*Ubr5^{mt}* ankle joints. (E,F) Whole mount β-Gal staining of (E) control and (F) pCAGG-*Ubr5^{mt}* ankle joints reveals β-gal expression in muscles and associate tendons. Sagittal section of ankle joint (G) showing an ectopic structure associated with the AT midbody (closed arrowhead) stained positive for *Ubr5*/*UBR5* expression.
(TIF)

**S3 Fig. UBR5 and PPS levels correlate with human AC damage.** (A-C) Examples of human OA patient material stained with (A) haematoxylin and eosin (H&E), (B) toluidine blue or (C) safranin O revealed intra- and inter-sample variation in AC defects. Coloured boxes indicate regions of the varying OA severity (please see figure key). (A-C) Colour-coded, magnified dashed boxes in upper panels are shown in more detail in the colour-coded lower panels (thick outlines). Moderate-scored regions (orange) exhibited extensive surface fibrillation and reduced toluidine blue and safranin O staining in comparison to low-scored regions (green). Severe-scored regions (red) exhibited loss of safranin O staining and apical-basal clefts in the AC surface. (C) The dashed black lines indicate the apical edge of the AC. (D-K) Human AC samples graded as low, mild or moderately damaged were analysed for (D,F,H) PKA activity (PPS) and (E,G,I) UBR5 expression. Graphs of percentage of (J) PPS and (K) UBR5 positive

cells for low and combined values for mild and moderate AC grades. Mean and s.e.m indicated. n = six biological replicates. Fishers exact test on pooled cell count data. p = <0.0001 for both.
(TIF)

**S4 Fig. Spectrum of *Ubr5^mt* and *Ubr5^mt*+Smo^LoF associated tissue-specific metaplastic responses.** Overview of the metaplastic events of various *Ubr5^mt* tissues. Red = non-cartilaginous tissues (Synovium, AT and Superficial Digital Flexor tendon); Orange = cartilaginous tissues (retinaculum); Yellow = calcified cartilage; Green = normotopic bone; Blue = heterotopic or enlarged normotopic bone. Arrows indicate the direction of metaplasia, with the arrowhead indicating the tissue type in 24-week-old *Ubr5^mt* and/or *Ubr5^mt*+Smo^LoF animals. Metaplastic tissue events unique to *Ubr5^mt*+Smo^LoF are indicated by an asterisk.
(TIF)

## Acknowledgments

We would like to thank Lorraine Rose for preliminary μCT scanning and the SURF and IGMM Histology facilities for their services. We also thank the BRF for expert technical assistance

## Author Contributions

**Conceptualization:** Katherine A. Staines, Colin Farquharson, Natalia A. Riobo-Del Galdo, Robert E. Hill, Mark Ditzel.

**Data curation:** Mark Ditzel.

**Formal analysis:** Mark Ditzel.

**Funding acquisition:** Natalia A. Riobo-Del Galdo, Robert E. Hill.

**Investigation:** David Mellis, Katherine A. Staines, Silvia Peluso, Ioanna Ch. Georgiou, Natalie Dora, Malgorzata Kubiak, Rob van't Hof, Michela Grillo, Elaine Kinsella, Stuart H. Ralston, Donald M. Salter, Natalia A. Riobo-Del Galdo, Robert E. Hill, Mark Ditzel.

**Methodology:** David Mellis, Katherine A. Staines, Silvia Peluso, Natalie Dora, Rob van't Hof, Elaine Kinsella, Natalia A. Riobo-Del Galdo, Robert E. Hill, Mark Ditzel.

**Project administration:** Robert E. Hill.

**Resources:** Colin Farquharson, Anna Thornburn, Stuart H. Ralston, Donald M. Salter, Natalia A. Riobo-Del Galdo, Robert E. Hill.

**Supervision:** David Mellis, Katherine A. Staines, Natalie Dora, Robert E. Hill, Mark Ditzel.

**Visualization:** Mark Ditzel.

**Writing – original draft:** Robert E. Hill, Mark Ditzel.

**Writing – review & editing:** David Mellis, Katherine A. Staines, Silvia Peluso, Colin Farquharson, Elaine Kinsella, Stuart H. Ralston, Donald M. Salter, Natalia A. Riobo-Del Galdo.

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
