## [Decision Letter · Decision Letter 0]

11 Feb 2021

Dear Dr Hill,

Thank you very much for submitting your Research Article entitled 'Ubiquitin-protein ligase Ubr5 cooperates with Hedgehog signalling to promote skeletal tissue homeostasis' to PLOS Genetics.

The manuscript was fully evaluated at the editorial level and by three peer reviewers. As you will see, all reviewers are enthusiastic, as are the editors. Based on the review comments and our editorial evaluation, we ask that you prepare a revised version with changes to the presentation to address the reviewers' concerns. We do not think major changes in organization are necessary. 

Your revisions should address the specific points made by each reviewer.

[LINK]

Yours sincerely,

Gregory S. Barsh

Editor-in-Chief

PLOS Genetics

Gregory Copenhaver

Editor-in-Chief

PLOS Genetics

Reviewer's Responses to Questions

**Comments to the Authors:**

Reviewer #1: attached

Reviewer #2: This is a large amount of work, much of which has been conducted in vivo, in very technically challenging tissues.

The primary aim of the manuscript is to explore the skeletal influence of UBR5, a regulator of the hedgehog pathway. The manuscript describes this and proposes possibilities concerning molecular mechanisms. The authors have previously linked, in a series of papers, this ligase to hedgehog signalling including the regulation of hedgehog ligand expression in the limb where effects on tissue development were minimal.

There is no question that increasing our understand of the regulation of Hh signalling in the mammalian, particularly mature, skeleton is a worthy question as there is increasing confusion particularly regarding the balance of canonical (which has so far received more attention) and non-canonical Hh roles. The authors frame this nicely in the introduction.

General comments

In summary I would say this manuscript adds knowledge to this area, its, perhaps apparently paradoxical and at times slightly superficial and descriptive findings, are worthy of publication. It is extremely important that all information, including that which is mechanistically unresolved or at apparent odds with dogmas, is in the research domain.

Fundamentally there is multi-faceted analysis, sometimes unnecessarily quantitated, to support the major descriptive findings in the mouse backed up by follow-up in vitro studies. Importantly, I am in agreement with the central tenets of the manuscript as summarised, they are supported by the evidence, but I feel that for maximal impact the manuscripts readability could be markedly improved.

This is often points of clarity, improvements to figures and legends etc, but also the authors may wish to consider the salient findings and decluttering the main figures- as in current form the paper is very dense and not easy to follow. The authors are to be commended for a valiant attempt to mechanistically explain the mechanistic links to Hh and PKA signalling in vitro which add support to interpretations of the findings.

As the authors note it is of particular interest that the phenotypes are only apparent in maturing or mature context despite the developmental nature of the pertubation. What do the authors propose might be the reason for this? Does an evolving landscape of Hh signalling account for this and/or are their mechanical inputs into this regulatory axis? Perhaps linkable to locations of ectopic ossification/ failure of cartilage at interface with bone (tears).

I wasn’t clear throughout – are the effects limited to hind limbs? Do you have images of forelimbs?

Was cartilage thickness measured- were effects bias to compartments ? Is cartilage loss a consequence of changes to other soft tissues in limb? Changes to gait – do mice move apparently normally (beyond squat resting position).

Were mice weights normal ? Perhaps previously described.

What were genders of small groups (n=3)

Minors:

I found Figure 1 early panels extremely hard to navigate. Line 108 are ankles shown?

Line 120 authors conveying this shows us where UBR5 is ? - consider re-phrase plain english

Lines 123 beta-gal activity, thus they are derived from embryonic mesenchyme ?

Line 134 different to what?

Lines 135-137 - no comparative controls to aid reader to see 'ectopic' - danger the non-specialist reader is completely lost

Line 145 “joints to reveal”

Concern over n throughout in vivo studies (Limitations should be noted in discussion). No control for cage to cage variability possible with such low n. N could be more easily found for some studies too (key result descriptor- see ARRIVE guidelines). N is often 3 animals ?

Legends are, however, detailed throughout suggesting good accordance with ARRIVE.

At times completely unclear as to n (e.g supplementary Figure 1 legend concerning gender breakdown for C).

Figures – often at times magnifications are not matched (Fig 3H/I) , controls are missing, more labelling would help reader. Quantitation does not immediately tally with images (e.g high density regions in Fig 3B-F, and quantitation in G – 10-15% red?)

K is an example of unnecessary quantitation as also seen in Fig 5G ?

Fig 4 N and O – please label up – what are we looking for? How was the data for 4Q generated?

Commend authors on correct use of Fisher's tests for contingency data.

Reviewer #3: In this manuscript by Mellis and colleagues, they do a very robust characterisation of the role of UBR5 in skeletal homeostasis. Through genetic deletion of UBR5 they show that mice develop profound and progressive cartillage degradation and then metaplasia. Mechanistically, given the activation of GLI following UBR5, suprisingly suppressing smoothened exacerbated the phenotype providing potentially interesting novel mechanism. I think better integration of in vitro mechanism and in vivo phenotypes would improve the paper. However, i think this is an interesting phenotype that is worthy of publication.

Specific comments:

1) I think the phenotyping done in here is excellent and Im happy with this analysis of the paper

2) As the mechanistic analysis is surprising eg the fact that smoothened loss exarcerbated but did not suppress the phenotype, the key i think is how the loss of function of smoothened affected the gli induction.. It is clear that GLI is up in the UBR5 deleting animals so does Smoothened loss inhibit this. This need to be quantified (eg similar to work in figure 7)

3) in terms of the mechanism, have the authors looked at forskolin in vivo?

4) Likewise are the changes in any of the proposed pathways in figure 8E altered in the tissue e.g BRD4 etc...? It would help if the author could integrate their findings 3T3 cells with their in vivo tissue analysis.

**Have all data underlying the figures and results presented in the manuscript been provided?**

Reviewer #1: Yes

Reviewer #2: Yes

Reviewer #3: None

PLOS authors have the option to publish the peer review history of their article (what does this mean?). If published, this will include your full peer review and any attached files.

Reviewer #1: No

Reviewer #2: No

Reviewer #3: No

---

## [Editor Report · Decision Letter 1]

20 Mar 2021

Dear Dr Hill,

We are pleased to inform you that your manuscript entitled "Ubiquitin-protein ligase Ubr5 cooperates with hedgehog signalling to promote skeletal tissue homeostasis" has been editorially accepted for publication in PLOS Genetics. Congratulations!

The revised manuscript was evaluated by members of the editorial board; there is a consensus that the changes to organization and presentation have addressed the concerns raised at the initial round of review.

Yours sincerely,

Gregory S. Barsh

Editor-in-Chief

PLOS Genetics

Gregory Copenhaver

Editor-in-Chief

PLOS Genetics

Comments from the reviewers (if applicable):

**Data Deposition**

http://datadryad.org/submit?journalID=pgenetics&manu=PGENETICS-D-20-01820R1

**Press Queries**

---

## [Editor Report · Acceptance letter]

31 Mar 2021

PGENETICS-D-20-01820R1 

Ubiquitin-protein ligase Ubr5 cooperates with hedgehog signalling to promote skeletal tissue homeostasis 

Dear Dr Hill, 

We are pleased to inform you that your manuscript entitled "Ubiquitin-protein ligase Ubr5 cooperates with hedgehog signalling to promote skeletal tissue homeostasis" has been formally accepted for publication in PLOS Genetics! Your manuscript is now with our production department and you will be notified of the publication date in due course.

With kind regards,

Andrea Szabo

PLOS Genetics

On behalf of:
